# One reaction to make highly stretchable or extremely soft silicone elastomers from easily available materials

Pengpeng Hu [1], Jeppe Madsen[1] & Anne Ladegaard Skov [1]✉

Highly stretchable, soft silicone elastomers are of great interest for the fabrication of stretchable, soft devices. However, there is a lack of available chemistries capable of efficiently preparing silicone elastomers with superior stretchability and softness. Here we show an easy curing reaction to prepare silicone elastomers, in which a platinum-catalyzed reaction of telechelic/multi-hydrosilane (Si–H) functional polydimethylsiloxane (PDMS) in the presence of oxygen and water leads to slow crosslinking. This curing chemistry allows versatile tailoring of elastomer properties, which exceed their intrinsic limitations. Specifically, both highly stretchable silicone elastomers (maximum strain of 2800%) and extremely soft silicone elastomers (lowest shear modulus of 1.2 kPa) are prepared by creating highly entangled elastomers and bottle-brush elastomers from commercial precursor polymers, respectively.

---

[1] Danish Polymer Centre, Department of Chemical and Biochemical Engineering, Technical University of Denmark, DTU, Søltofts Plads, Building 227, 2800 Kgs Lyngby, Denmark. ✉email: al@kt.dtu.dk

**H**ighly stretchable, soft silicone elastomers are of great interest for the fabrication of stretchable electronics, soft actuators, medical devices, and microfluidics[1–5]. High stretchability provides long-term device stability in various distortion scenarios and permits exceptional deformations. Significant effort has been devoted to preparing silicone elastomers with a combined softness and elasticity resembling that of human soft tissue for use in soft robotics[6,7].

Silicone elastomers are typically prepared by crosslinking linear polymers with cross-linkers. Figure 1a presents one of the most commonly used crosslinking reactions—i.e., the hydrosilylation reaction of telechelic vinyl functional polydimethylsiloxane (PDMS) with a multi-hydrosilane (Si–H) functional cross-linker in the presence of a platinum catalyst[8]. Based on classical curing chemistry, network strands possess the same size and structure as the precursor polymers, which ultimately determine the mechanical properties of the silicone elastomers. The ultimate

extensibility of the resulting silicone elastomers is proportional to $M^{0.5}$ based on the Kuhn model, where $M$ is the average molar mass of network strands[9]. The ultimate extensibility is usually less than 900%, and it is difficult to further increase this value by using longer precursor polymers, since they would bring difficulties to the fabrication process due to their high viscosity[10]. The elastic modulus of the silicone elastomers is determined by the crosslinking density—namely, the molar density of mechanically active strands $\nu = \rho/M$, where $\rho$ is the density. However, the lowest achievable elastic modulus for ideal elastomer networks is around 0.6 MPa due to the fact that entanglements in the crosslinked networks act as topological crosslinks once the molecular weight of network strands ($M$) exceeds the entanglement molecular weight[11].

Several strategies have been explored for overcoming limitations on the ultimate extensibility and softness of silicone elastomers. For example, elastomers prepared from long precursor polymers in

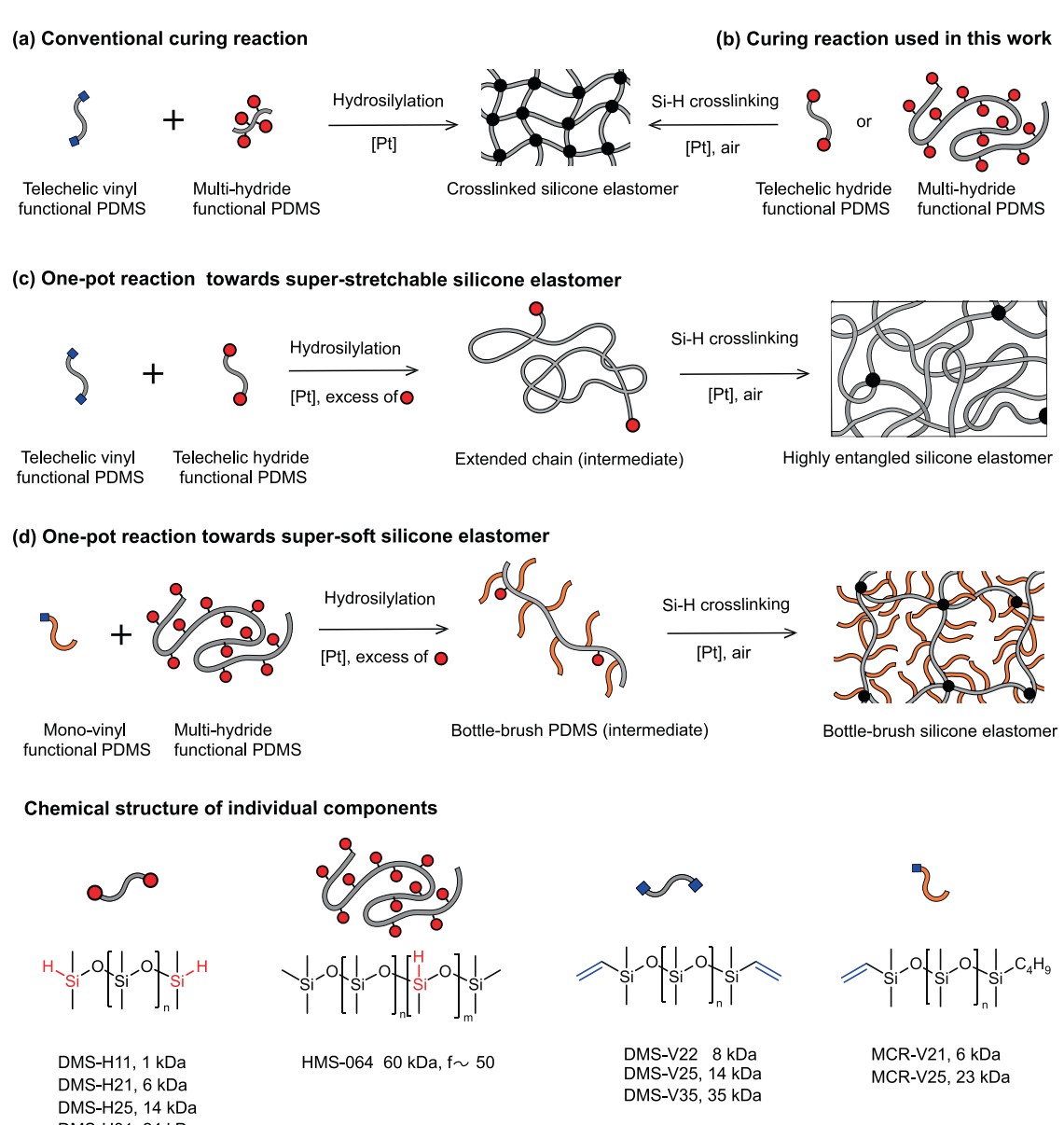

**(a) Conventional curing reaction**

Telechelic vinyl functional PDMS + Multi-hydride functional PDMS → Hydrosilylation [Pt] → Crosslinked silicone elastomer ← Si-H crosslinking [Pt], air → Telechelic hydride functional PDMS or Multi-hydride functional PDMS

**(b) Curing reaction used in this work**

**(c) One-pot reaction towards super-stretchable silicone elastomer**

Telechelic vinyl functional PDMS + Telechelic hydride functional PDMS → Hydrosilylation [Pt], excess of ● → Extended chain (intermediate) → Si-H crosslinking [Pt], air → Highly entangled silicone elastomer

**(d) One-pot reaction towards super-soft silicone elastomer**

Mono-vinyl functional PDMS + Multi-hydride functional PDMS → Hydrosilylation [Pt], excess of ● → Bottle-brush PDMS (intermediate) → Si-H crosslinking [Pt], air → Bottle-brush silicone elastomer

**Chemical structure of individual components**

DMS-H11, 1 kDa
DMS-H21, 6 kDa
DMS-H25, 14 kDa
DMS-H31, 24 kDa

HMS-064 60 kDa, f∼ 50

DMS-V22   8 kDa
DMS-V25, 14 kDa
DMS-V35, 35 kDa

MCR-V21, 6 kDa
MCR-V25, 23 kDa

Telechelic hydride functional PDMS    Multi-hydride functional PDMS    Telechelic vinyl functional PDMS    Mono-vinyl functional PDMS

**Fig. 1 Various curing reactions to prepare silicone elastomers. a** Conventional curing reaction to prepare silicone elastomers.[8] **b** Curing reaction used in this study to prepare silicone elastomers. **c** Highly stretchable silicone elastomers and **d** extremely soft silicone elastomers prepared from a one-pot curing reaction by combining hydrosilylation and subsequent Si–H crosslinking reactions.

solution and subjected to solvent evaporation after curing, supra-molecular elastomers with movable cross-links, and concatenated ring elastomers have all been developed to improve ultimate extensibility[12–17]. Approaches such as adding external sol molecules, sparse crosslinking, and crosslinking bottle-brush PDMS have also been used to prepare soft silicone elastomers[18–23]. While the above strategies improve elastomer extensibility or softness to some extent, they either increase process complexity or lead to mechanical instabilities in the resulting elastomers. What is more, none of them are versatile enough to enable the preparation of silicone elastomers that are both highly stretchable and very soft. There is thus a lack of available chemistries capable of efficiently preparing silicone elastomers with superior stretchability and softness.

Silicone elastomers have been prepared from Si–H functionalized polymers at 250 °C in air, where the crosslinking mechanism was found to originate from oxidative crosslinking of Si–H groups in the presence of oxygen[24]. In this study, silicone elastomers are prepared from PDMS with telechelic/multiple Si–H groups and a platinum catalyst in air at the much lower temperature of 100 °C (see Fig. 1b). In contrast to classical curing chemistries where the network strands are directly related to the length of the precursor polymers, the curing chemistry presented here, when combined with the hydrosilylation reaction, allows network strands to be tailored from normal linear precursors during the curing process. Specifically, we report highly stretchable silicone elastomers with ultra-long network strands and extremely soft silicone elastomers with bottle-brush strands, both of which are easily prepared from commercially available linear precursors. Both silicone elastomers are based on sequential crosslinking mechanisms in one-pot reactions, where the fast hydrosilylation reaction is followed by a slow cross-linking of residual Si–H functional groups. This allows independent control of network strand size and structure, as well as of crosslinking.

## Results

**Elucidation of curing reaction.** In our previous work with platinum-catalyzed hydrosilylation reactions, we observed that elastomers were formed when only telechelic Si–H-functional PDMS and a platinum catalyst were present. This discovery of a potentially simple method for preparing elastomers prompted us to perform further studies in order to elucidate the mechanism of this formation.

As well-known side reactions of the conventional curing reaction (Fig. 1a), hydrolysis of hydrosilanes with atmospheric moisture, and oxidation of hydrosilanes with atmospheric oxygen both lead to the formation of Si–OH groups[25,26]. The Si–OH groups can form Si–O–Si by reacting with themselves or with Si–H groups[27,28]. In the presence of a platinum catalyst, moisture, and oxygen, telechelic Si–H functional PDMS can thus be transferred into telechelic Si–OH, which may further undergo condensation to form extended chains. However, these reactions cannot account for the formation of elastomers from telechelic Si–H functional PDMS in a conventional manner. Instead, obtaining elastomers from telechelic Si–H functional PDMS requires either the formation of concatenated rings through intramolecular condensation reactions or some other unexplored crosslinking reaction[15,17]. Water and oxygen are the possible reactants in the curing reaction with Si–H-containing precursor PDMS. The reaction was therefore carried out under conditions in which water and oxygen content could be carefully controlled. Specifically, in a series of experiments, a representative telechelic Si–H functional PDMS (DMS-H11, $M_n = 1$ kDa) was heated at 100 °C for 48 h in a sealed flask under each of the following reaction conditions: dry $N_2$ (a dried sample under dried

nitrogen), wet $N_2$ (a sample containing water with ~4 molar equivalent of hydrosilanes under dried nitrogen), dry air (a dried sample under dried air), and wet air (a sample containing water with ~4 molar equivalent of hydrosilanes under dried air). DMS-H11 was converted into a solid elastomer only under wet air conditions. As measured by [1]H nuclear magnetic resonance (NMR), the conversion efficiency of Si–H functional groups for the three liquid products under dry $N_2$, wet $N_2$, and dry air conditions were found to be 3.3%, 32.8%, and 52.7%, respectively (Fig. 2a). These findings indicate that both water and oxygen participate in the reaction. A peak at 2.28 ppm in the [1]H NMR spectra of the samples from the wet $N_2$ and dry air atmospheres is assigned to a Si–OH structure (Fig. 2a), suggesting the formation of Si–OH from Si–H with water and oxygen, respectively[29,30]. A further condensation process of Si–OH may result in chain extension, as evidenced by the increased molecular weight of the samples (Supplementary Fig. 1). Another new peak that appears at 3.49 ppm (Fig. 2a) on the [1]H spectrum of the sample from dry air atmosphere is contributed to a silyl ether (Si–O–$CH_2$–Si) structure which has been reported to originate from oxidation reactions of Si–H with oxygen and methyl groups at much higher temperatures in the absence of a platinum catalyst[24]. The presence of this silyl ether suggests that branched chains are formed during the oxidation of Si–H. The integration of the [1]H spectrum shows that the amount of hydrogen on silyl ether and Si–OH only accounts for 5.5% of the Si–H loss (Supplementary Table 1), suggesting that most Si–H are reacted into other structures which are not identified in the [1]H spectrum. The minimal loss of Si–H (3.3%, Fig. 2a) under dry $N_2$ conditions is most likely due to trace amounts of air and water in the starting polymer, as well as a small amount of vinyl groups associated with the platinum catalyst.

Direct investigations of the chemical composition of a precursor polymer DMS-H11and its reaction products were performed using [29]Si NMR. Figure 2b shows that a peak at −7.0 ppm on the spectrum of the precursor polymer is assigned to Si–H functional groups[31]. This peak vanishes on the spectrum of the elastomer (Ela_DMS-H11), suggesting an efficient conversion of Si–H after curing. New peaks located at −37.6 ppm, −20.6 ppm, and 7.0 ppm are observed on spectra of the samples from wet $N_2$ and dry air atmospheres and are assigned to $CH_3SiHO_2$, $(CH_3)_2SiO_2$ adjacent to $CH_3SiHO_2$, and $(CH_3)_3SiO$(or $(CH_3)_2CH_2SiO$), respectively[31–33]. The peak at −37.6 ppm disappears, while the peak at 7.0 ppm remains and another new peak at −64.1 ppm occurs on the spectrum of the elastomer. The new peak at −64.1 ppm is attributed to $CH_3SiO_3$[31], which could be final structure after the hydrolysis and the further condensation reactions of the Si–H functional group on $CH_3SiHO_2$.

The hydrolysis of Si–H and further condensation of Si–H with Si–OH group would only produce more backbone chains (–$(CH_3)_2SiO$–), which cannot explain the newly formed structures mentioned above. Integration of the spectra of the elastomers shows that Si atoms associated with these newly formed structures of $(CH_3)_3SiO$ (or $(CH_3)_2CH_2SiO$) and $CH_3SiO_3$ account for 1.5–2.5% of Si atoms in the elastomers (Supplementary Table 2), suggesting that these structures play a major role in crosslinking.

Based on the observations above, the oxidation reactions of Si–H functional groups with water and oxygen to produce the newly formed structures are proposed as follows. Si–H functional groups are turned into silyl radicals (O$(CH_3)_2$Si·) through hydrogen abstraction in the presence of oxygen or water (Fig. 2c-A and E)[34,35]. After further oxidation processes with oxygen, the silyl radicals are transformed into O$(CH_3)_2$SiO· (Fig. 2c-B and C)[24]. Oxidation processes also happen on methyl groups at ends of PDMS

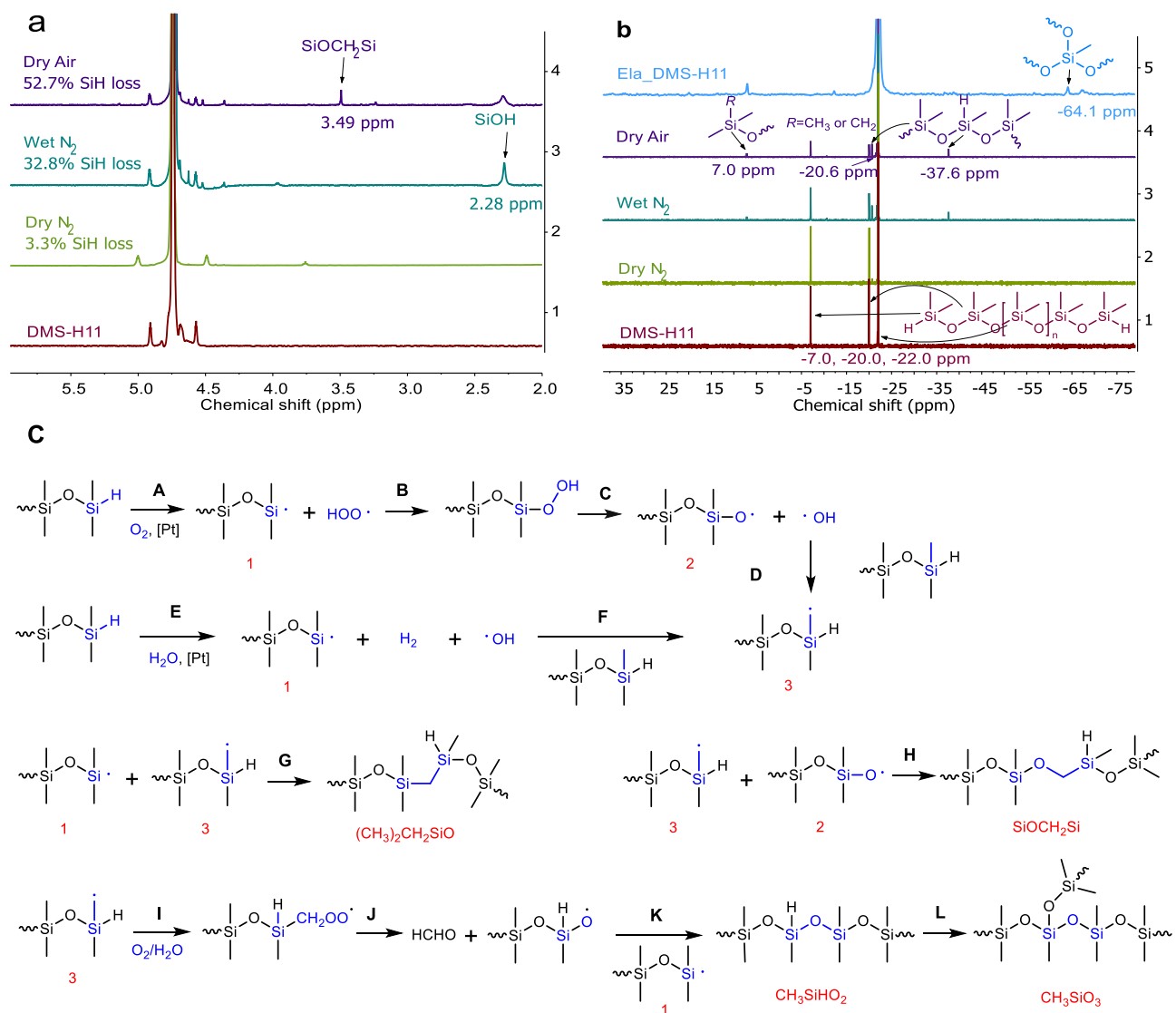

**Fig. 2 Mechanism studies of crosslinking of a telechelic Si–H functional PDMS. a** [1]H NMR spectra and **b** [29]Si NMR spectra of a telechelic Si–H functional PDMS (DMS-H11) and its reaction products after heating at 100 °C for 48 h under dry $N_2$, wet $N_2$, and dry air conditions, respectively. Additionally, [29]Si solid-state NMR spectrum of the elastomer (Ela_DMS-H11) cured under normal air atmosphere is shown in (**b**). **c** Proposed mechanism of crosslinking Si–H functional groups in the presence of oxygen, water, and a catalyst.

chains: oxygen initially generates a carbon radical ($SiCH_2\cdot$) by hydrogen atom abstraction (Fig. 2c-D and F)[36,37]. Branching structures like $(CH_3)_2CH_2SiO$ and $SiOCH_2Si$ are formed by combinations of $SiCH_2\cdot$ with $O(CH_3)_2Si\cdot$ and $O(CH_3)_2SiO\cdot$, respectively (Fig. 2c-G and H). Further oxidation of $SiCH_2\cdot$ leads to the creation of $SiCH_2OO\cdot$, which enables the formation of $OCH_3HSiO\cdot$, followed by the branching the structure of $CH_3SiO_3$ (Fig. 2c-I, J, K and L)[36,37].

It should be noted that the same radicals mentioned above were also used to explain crosslinking during the oxidization of Si–H functional PDMS at high temperature (250 °C) in the absence of a catalyst, in which a silyl ether (Si–O–$CH_2$–Si) structure was the main oxidized structure produced[24]. In our work, however, the oxidation process is faster and occurs at a much lower temperature due to the use of a platinum catalyst; the main oxidized structures in the elastomers are $(CH_3)_2CH_2SiO$ and $CH_3SiO_3$. Si–[Pt]–H complex is known to form by oxidative addition[38–40], and [Pt]–oxygen complex has been reported to significantly promote platinum catalyzed hydrosilylation reactions[38,39]. These compounds may allow the reaction with

oxygen and water to occur at moderate temperatures, thereby changing the main oxidized structures produced.

This curing chemistry can be extended to the curing of multifunctional Si–H functional PDMS, where both hydrolysis/ condensation and oxidation reactions may contribute to crosslinking. The prepared elastomers listed in Table 1 are named according to the precursor polymer used. Overall, the prepared silicone elastomers show tensile strains of 70–360%, Young's moduli of 0.3–0.6 MPa, and tensile strengths of 0.2–0.6 MPa. These mechanical properties are comparable to those of the two conventional silicone elastomers prepared via the classical curing route.[10] The platinum catalysts (Karstedt's catalyst and Speier's catalyst) and tris(dibutylsulfide) rhodium trichloride catalyst are shown as effective catalysts for the curing of telechelic Si–H functional DPMS (Supplementary Fig. 3c). The curing rate can be significantly improved by applying a lower hydride concentration of precursor polymer, higher temperature, and higher platinum catalyst concentration (Supplementary Fig. 3b and c). Particularly, a solid elastomer was formed from DMS-H25 ($M_n$ = 14 kDa, 0.14 mol/kg Si–H functional groups) after only 0.5 h under conditions of 150 °C and 30 ppm Karstedt's catalyst.

**Table 1 Silicone elastomers prepared from telechelic/multi Si–H functional PDMS in presence of a platinum-divinyl tetramethyldisiloxane complex.**

| Sample | Precursor length (kDa) | Tensile strain (%) | Tensile strength (MPa) | Young´s modulus (MPa) |
|---|---|---|---|---|
| Ela_DMS-H11 | 1 | 71 | 0.24 | 0.37 |
| Ela_DMS-H21 | 6 | 70 | 0.35 | 0.60 |
| Ela_DMS-H25 | 14 | 235 | 0.49 | 0.40 |
| Ela_DMS-H31 | 24 | 363 | 0.46 | 0.29 |
| Ela_HMS-064 | 60 | 118 | 0.52 | 0.52 |
| Ref_DMS-V25 | 14 | 127 | 0.54 | 0.84 |
| Ref_DMS-V41 | 35 | 482 | 0.44 | 0.19 |

Conventional elastomers (Ref_DMS-V25 and Ref_DMS-V41) were prepared as references from telechelic vinyl functional PDMS, a multi-Si–H functional PDMS, and a platinum catalyst.

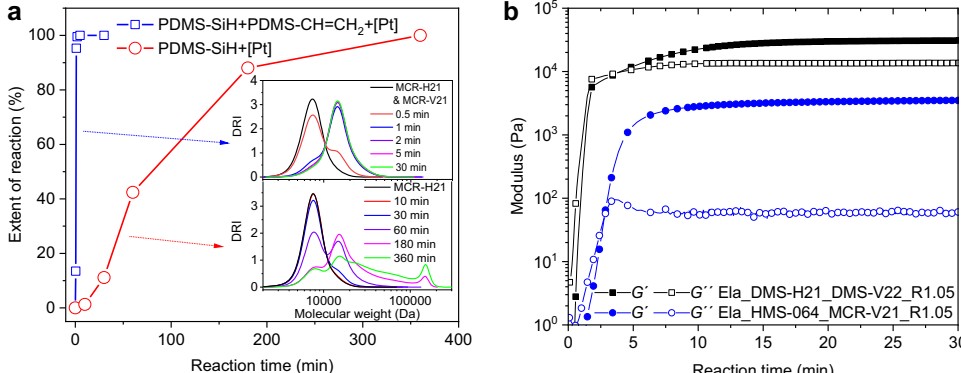

**Fig. 3 Kinetic studies of investigated reactions. a** Reaction progression over time for the platinum-catalyzed reactions at 100 °C of mono-hydrosilane functional PDMS, and mono-hydrosilane functional PDMS with mono-vinyl functional PDMS (stoichiometric amounts of hydride and vinyl). The extents of reaction were determined based on $^1$H NMR spectra (Supplementary Fig. 4). The inserts show molecular weight distributions over time during the two reactions. **b** Dependence of storage and loss moduli ($G´$ and $G´´$) of representative highly stretchable and extremely soft silicone elastomers on curing time measured at 100 °C, 1 Hz, and with a fixed strain of 1.0%.

In order to compare the reaction kinetics of the Si–H functional PDMS system with those of the hydrosilylation reaction between Si–H and vinyl groups, platinum-catalyzed reactions of mono-Si–H functional PDMS and mono-Si–H functional PDMS with mono-vinyl functional PDMS, respectively, were conducted at 100 °C. The total concentration of functional groups was the same for both reactions. Figure 3a shows that the reaction of mono-Si–H functional PDMS takes 6 h to complete 100%, compared to 2 min for the hydrosilylation reaction. In addition, the reaction of mono-Si–H functional PDMS products a fraction of products that are more than 10-time molecular weight of precursor polymers (Fig. 3a). This is consistent with the branching nature of Si–H oxidation. In comparison, the hydrosilylation reaction of mono-Si–H functional PDMS with mono-vinyl functional PDMS exclusively produces chains with double initial molecular weight (Fig. 3a).

**Curing chemistries for highly stretchable/extremely soft silicone elastomers**. Preparing silicone elastomers with long strands is one way to efficiently obtain high stretchability. However, doing so using conventional curing chemistry requires exceptionally long precursor polymers whose high viscosity brings difficulties to fabrication process[9,41]. Here, highly stretchable silicone elastomers are prepared from a platinum-catalyzed reaction between telechelic Si–H functional PDMS and telechelic vinyl functional PDMS using a small excess of Si–H groups (Fig. 1c). In a one-pot reaction, both the hydrosilylation reaction between Si–H and vinyl groups and the water and oxygen-mediated crosslinking of Si–H (see above) take place. Due to the significant kinetic advantage of the hydrosilylation reaction over Si–H crosslinking (Fig. 3a), the hydrosilylation

reaction is expected to proceed to high conversion before any significant crosslinking occurs. Assuming the two reactions happen strictly in sequence, the hydrosilylation reaction results in extended chains which are subsequently cross-linked into elastomers through the reaction of excess Si–H with oxygen and water. In this case, the average molar mass of the network strands can be expressed using Eq. (1) (derived from Supplementary Equation (1)):

$$M_{\text{extended}} = \frac{RM_{\text{DMS-H}} + M_{\text{DMS-V}}}{R - 1} \qquad (1)$$

where $M_{\text{DMS-H}}$ is the molecular weight of telechelic Si–H functional PDMS, $M_{\text{DMS-V}}$ is the molecular weight of telechelic vinyl functional PDMS, and $R$ is the molar ratio of the Si–H-to-vinyl functional groups. According to Eq. (1), even when the values of $M_{\text{DMS-H}}$ and $M_{\text{DMS-V}}$ are relatively low, ultra-long network strands can be achieved by setting $R$ close to unity.

Expanding the diameter of the polymer by attaching polymer brushes is a known method for diluting entanglements without markedly increasing chain stiffness. The resulting bottle-brush elastomers are intrinsically soft, with shear moduli of 1–100 kPa[21–23,42]. However, preparing bottle-brush elastomers generally involves relatively complex multistep syntheses, including preparation of bottle-brush polymers followed by subsequent crosslinking reactions[23,42]. Here, soft silicone elastomers are obtained by preparing bottle-brush elastomers through a platinum-catalyzed, one-pot curing reaction of multi-Si–H functional PDMS with mono-vinyl functional PDMS in the presence of excess of Si–H functional groups (Fig. 1d). During the reaction, bottle-brush polymers are preferentially formed by grafting mono-vinyl functional PDMS onto the multi-Si–H functional PDMS through hydrosilylation. The resulting bottle-brush polymers are

subsequently cross-linked into bottle-brush elastomers through the relatively slow crosslinking of Si–H in the presence of oxygen and water. Side chain lengths, and thus polymer diameter, are governed by the length of the mono-vinyl functional PDMS. Assuming crosslinking of Si–H takes place strictly after full side chain grafting, the molecular weight between Si–H groups on the bottle-brush chains can be determined as:

$$M_{c\_SiH} = \frac{M_{\text{brush}}}{f_{\text{brush}}} + 1 \qquad (2)$$

where $M_{\text{brush}}$ (Supplementary Equation (3)) and $f_{\text{brush}}$ (Supplementary Equation (4)) are the molecular weight and number of excess hydrides per chain, respectively, after full side chain grafting.

Representative curing reactions for preparing highly stretchable, extremely soft elastomers were investigated by tracing the evolution of the storage and loss moduli during the two curing reactions (Fig. 3b). Gel points are reached within 5 min at 100 °C, suggesting fast curing processes.

**Properties of highly stretchable silicone elastomers**. A number of stretchable silicone elastomers were prepared using different hydrosilane-to-vinyl-functional polymer ratios as well as polymers of different molecular weights, as shown in Table 2. Figure 4a shows that, when using the same precursor polymers DMS-H21 and DMS-V22, the tensile strain increases from 1040 to 2400% when $R$ decreases from 1.15 to 1.05. Tensile strain can be further increased to 2800% by using longer starting polymers DMS-H25 and DMS-V25. Longer extended chains improve tensile strain by enabling larger slippage lengths upon deformation (Table 2). The Kuhn model is widely used to estimate the ultimate extensibility of elastomers as $\lambda_{\text{max}} = L/h$, where $L$ is the strand length in a fully stretched state, and $h$ is the strand length in a random coil state[9]. $\lambda_{\text{max}}$ is thus proportional to $M^{0.5}$ based on the relations of $L \propto M$ and $h \propto M^{0.5}$. A linear relation between $\lambda_{\text{max}}$ and $M_{\text{theo}}^{0.5}$ complies with the Kuhn model (Supplementary Fig. 5), suggesting that Eq. (1) is a valid description of the molar mass of the network strands, which supports the proposed curing route. Silicone elastomers with tailored stretchability can thus be realized by designing strand lengths based on Eq. (1). Elastomers' linear viscoelastic responses are shown in Fig. 4b. The storage modulus of the conventional elastomer reaches a plateau at low frequencies, while the storage moduli of the highly stretchable elastomers continue to decrease as the frequency approaches zero. This unusual behavior is explained by stress relaxation from entanglements of highly extended strands upon deformation. Stretchable elastomers are often biaxially stretched in practical use[43,44]. Figure 4c shows the area of stretchable silicone elastomer, Ela_DMS-H21_DMS-V22_R1.05, is biaxially extended 180-fold from an initial state—20 times greater extension than that of the conventional silicone elastomer Ref_DMS-V41. This significantly enhanced stretchability demonstrates the very high mechanical integrity of the elastomers studied here. Swelling experiments show 67–83% gel fractions of the elastomers (Supplementary Table 3). The most stretchable elastomer (Ela_DMS-H25_DMS-V25_R1.05) exhibits the highest swelling ratio of 128,

which is around 10 times larger than the conventional elastomer (Ref_DMS-V25). The extremely high swelling ratio correlates with the high flexibility of the elastomer where entanglements greatly outnumber crosslinks.

**Properties of extremely soft silicone elastomers**. A range of bottle-brush elastomers were fabricated according to the formulations in Table 3. Shear moduli of the prepared elastomers ($G$) are taken to be equivalent to the plateau values of storage moduli at low oscillatory frequency (Fig. 5a), and decrease from 7.4 kPa to 1.2 kPa when using either smaller $R$ or 4-fold longer side chains. Such low shear moduli are comparable to those of hydrogels and human soft tissue[45]. Linear viscoelastic responses (Fig. 5a) show that the prepared elastomers have near frequency-independent storage moduli, making them resemble a perfect rubber. Table 3 shows that the molecular weights of bottle-brush network strands ($M_c$) (Supplementary Equation (6)) are larger than the average molecular weight between Si–H groups on the bottle-brush chains ($M_{c\_SiH}$). Specially, $M_c > 10 M_{c\_SiH}$ for the elastomers with MCR-V21 side chains. The large differences between $M_c$ and $M_{c\_SiH}$ can be explained by the preferentially intramolecular reactions of the multi-functional bottle-brush chains, which result in a large fraction of elastically inactive loops and dangling[46]. This is illustrated by the lower gel fractions (52-83 %) compared to the conventional elastomer (97%), see Supplementary Table 3. Figure 5b shows the compressibility of the bottle-brush elastomer Ela_HMS-064_MCR-V21_R1.05 compared to that of the conventional elastomer Ref_DMS-V41. The bottle-brush elastomer is compressed to a strain of 88% under a pressure of 0.16 MPa, while the conventional elastomer only shows a compression strain of 19% under the same pressure. Importantly, despite the large compression strain imposed on the bottle-brush elastomer, it recovers to its initial state almost instantaneously upon pressure being released, displaying superior elasticity compared to normal soft elastomers, which often recover only partially.

**Discussion**

Silicone elastomers with bespoke properties can be prepared via classical hydrosilylation chemistry combined with a platinum-catalyzed reaction of telechelic/multi-hydride functional PDMS, without using any additional cross-linker. The mechanism of the curing reaction is consistent with platinum-mediated crosslinking of hydrosilanes in the presence of trace water and oxygen, and thus may be considered a side-reaction in conventional formulations. Compared with classical curing chemistry—i.e., hydrosilylation reaction—Si–H crosslinking in the presence of moisture and oxygen proceeds much more slowly, thereby providing formulations with an inherent delayed crosslinking opportunity and allowing the preparation of highly diverse networks using simple one-pot reactions.

Highly stretchable silicone elastomers and extremely soft silicone elastomers were developed by combining this curing chemistry with hydrosilylation reactions: the fast hydrosilylation reactions controlled the size and structures of network strands, after which elastomers were created through the much slower

**Table 2 Specifications for studied highly stretchable silicone elastomers.**

| Samples | $M_{\text{DMS-H}}$ (kDa) | $M_{\text{DMS-V}}$ (kDa) | $R$ | $M_{\text{extended}}$ (kDa) | $\lambda_{\text{max}}$(%) |
|---|---|---|---|---|---|
| Ela_DMS-H21_DMS-V22_R1.05 | 6 | 8 | 1.05 | 286 | 2493 |
| Ela_DMS-H21_DMS-V22_R1.10 | 6 | 8 | 1.10 | 146 | 1602 |
| Ela_DMS-H21_DMS-V22_R1.15 | 6 | 8 | 1.15 | 99 | 1142 |
| Ela_DMS-H25_DMS-V25_R1.05 | 14 | 14 | 1.05 | 574 | 2864 |

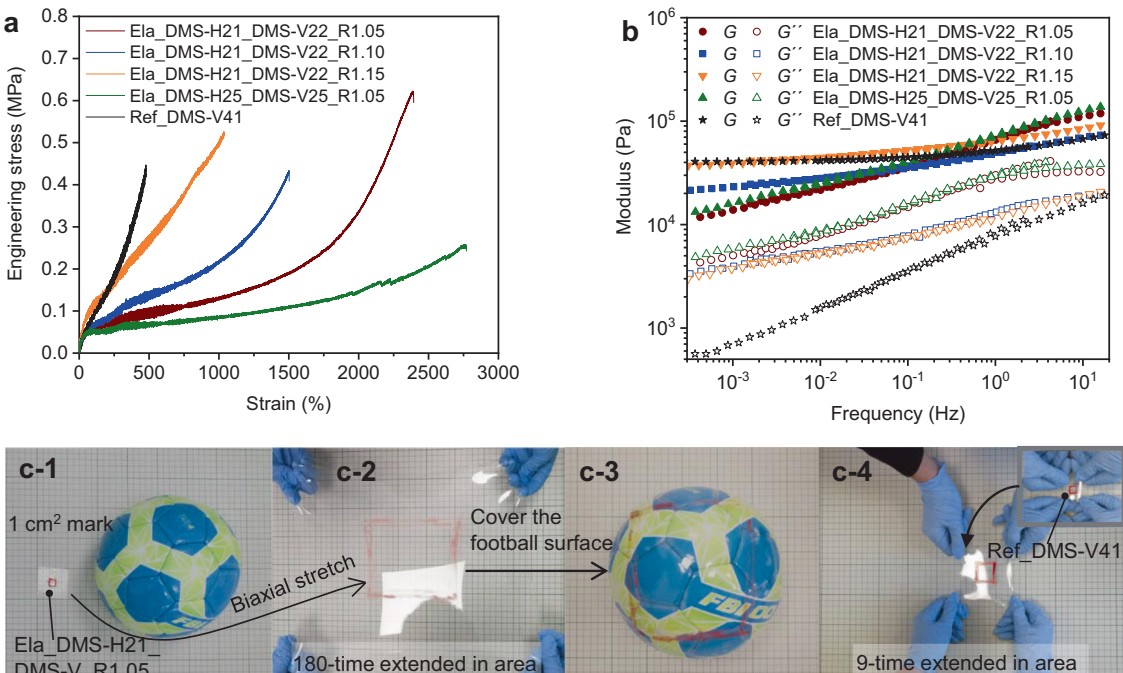

**Fig. 4 Properties of highly stretchable silicone elastomers and a conventional silicone elastomer. a** Uniaxial stress–strain curves. **b** Frequency dependence of storage and loss moduli ($G'$ and $G''$) measured at room temperature. **c** Biaxial stretching. (c-1) The highly stretchable silicone elastomer is marked with a red 1 cm$^2$ square; (c-2) the area of the same film is manually extended 180-fold. (c-3) The extended film subsequently covers the surface of a football with a diameter of 21 cm. (c-4) The conventional silicone elastomer marked with a 1 cm$^2$ red square is extended to a maximum of 9 times its original area.

**Table 3 Specifications for studied soft silicone elastomers.**

| Samples | R | $M_{HMS}$ (kDa) | $M_{MCR-V}$ (kDa) | $M_{c-SiH}$ (kDa) | $M_c$ (kDa) |
|---|---|---|---|---|---|
| Ela_HMS-064_MCR-V21_R1.05 | 1.05 | 60 | 6 | 130 | 631 |
| Ela_ HMS-064_MCR-V21_R1.20 | 1.20 | 60 | 6 | 42 | 525 |
| Ela_ HMS-064_MCR-V21_R1.50 | 1.50 | 60 | 6 | 18 | 224 |
| Ela_ HMS-064_MCR-V25_R1.05 | 1.05 | 60 | 23 | 342 | 543 |

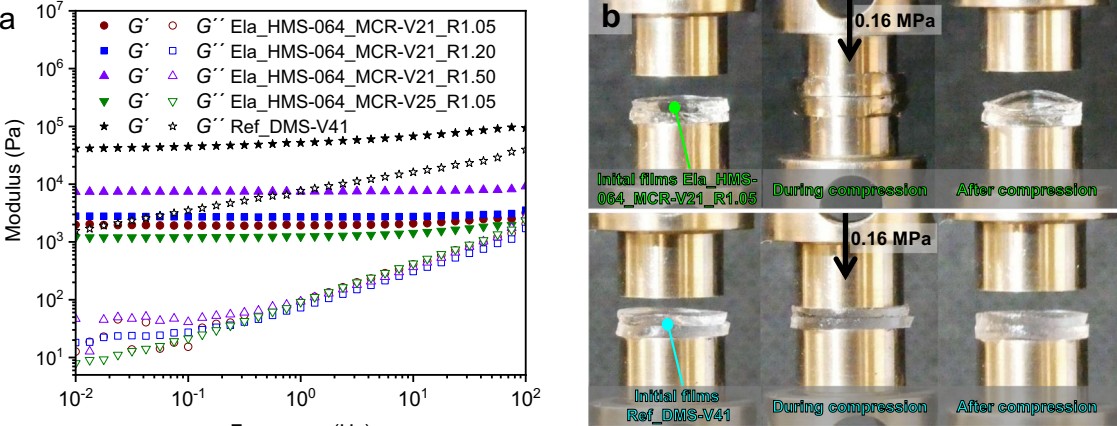

**Fig. 5 Properties of extremely soft silicone elastomers and a conventional silicone elastomer. a** Frequency dependence of storage and loss moduli ($G'$ and $G''$) of extremely soft silicone elastomers and a conventional silicone elastomer measured at room temperature. **b** Two stacked extremely soft specimens (8 mm in diameter) are compressed to a strain of 88% by a pressure of 0.16 MPa (the excess polymer is displaced up and down the sides of the geometries). After the pressure is released, the films are recovered from the compressed state. In comparison, two stacked conventional specimens (8 mm in diameter) were compressed to a strain of only 19% under the same pressure.

crosslinking of Si–H functional groups. Specifically, highly stretchable silicone elastomers were prepared by creating highly entangled (long-chain) silicone elastomers from the reaction between telechelic Si–H functional PDMS and telechelic vinyl functional PDMS. Tensile strains could be tailored from 1500% to 2800% by varying precursor length and the molar ratio of Si–H-to-vinyl groups. We demonstrated a 180-fold extension in area by biaxial stretch for one such highly stretchable silicone elastomer. Extremely soft silicone elastomers were made by creating bottle-brush silicone elastomers from the reaction between multi-Si–H functional PDMS and mono-vinyl functional PDMS. The shear moduli of the prepared bottle-brush elastomers could be adjusted from 1.2 kPa to 7.4 kPa by changing the molar ratio of reactive groups and the side chain lengths.

Both highly stretchable silicone elastomers and extremely soft silicone elastomers can be easily prepared via one-pot reactions using commercial precursors. In addition to enabling the preparation of highly stretchable or extremely soft elastomers, the general methodology based on slow crosslinking presented here enables the easy development of silicone elastomers with a wide range of functionalities.

## Methods

**Materials**. Information of polymers used in this studies is shown in Table 4, in which all the polymers were purchased from Gelest. Platinum-divinyl tetramethyl-disiloxane complex (SIP6830.3, 3.0 wt% Pt) was purchased from Gelest. Platinum cyclo-vinylmethyl siloxane complex (Catalyst 511, 1.0 wt% Pt) was purchased from Hanse Chemie. As a precursor polymer for condition controlling experiments, DMS-H11 (10 mL) was diluted in hexane (20 mL, ≥95%, Sigma-Aldrich) and dried with silica gel (5 g, particle size of 63–200 μm, high-purity grade, Sigma-Aldrich) in a sealed flask at a room temperature for 2 days. The upper layer was transferred into a dried flask through a syringe mounted with a filter. Subsequently, the hexane was thoroughly distilled under a vacuum pressure condition at room temperature for 6 h. Red pigment (PGRED01, 50% in silicone oil) was purchased from Gelest.

**Silicone elastomers prepared from hydrosilane (Si–H) containing PDMS**. All the telechelic Si–H functional PDMS and multi-Si–H functional PDMS in Table 4 were used to prepare silicone elastomers, respectively. A representative procedure is as follows: DMS-H11 (10 g, $1.00 \times 10^{-2}$ mol) was mixed with catalyst SIP 6830.3 (2 mg, $3.08 \times 10^{-7}$ mol) using a speed mixer (DAC150FVZ, Hauschild Co.) at 3000 rpm for 2 min. The mixture was poured into a mold and placed in an oven at 100 °C for 24 h. Selected elastomers were studied by $^{29}Si$ solid-state NMR.

**Highly stretchable silicone elastomers and extremely soft silicone elastomers**. Vinyl functional PDMS and catalyst SIP 6830.3 (2 mg, $3.08 \times 10^{-7}$ mol) were well-mixed using a speed mixer. Subsequently, Si–H functional PDMS was added into the mixture and well-mixed. The final mixture was poured on a mold and placed in an oven at 100 °C for 24 h. The prepared elastomers listed in Table 5 are named according to the precursor polymer used.

**Conventional silicone elastomers**. Part A and part B were prepared before the curing reaction. For a reference sample of Ref_DMS-V25, part A was prepared by mixing DMS-V25 (5 g, $3.3 \times 10^{-4}$ mol) with HMS-301 (0.69 g, $3.6 \times 10^{-4}$ mol). Part B was prepared by mixing DMS-V25 (5 g, $3.3 \times 10^{-4}$ mol) with catalyst 511 (2 mg, $1.0 \times 10^{-7}$ mol). For a reference sample of Ref_DMS-V41, part A was prepared by mixing DMS-V41 (10 g, $1.5 \times 10^{-4}$ mol) with HMS-301 (0.14 g, $7.3 \times 10^{-5}$ mol). Part B was prepared by mixing DMS-V41 (4.8 g, $7.7 \times 10^{-5}$ mol) with catalyst 511 (3 mg, $1.5 \times 10^{-7}$ mol). Parts A and B were then mixed together using a speed mixer at 3000 rpm for 30 s. The final mixture was poured on the surface of a polyethylene terephthalate (PET) substrate and evenly distributed by applying an automatic applicator. The PET substrate together with the mixture was placed in an oven at 100 °C for 5 h.

**Condition controlling experiments**. Before reactions, glassware was dried at 120 °C overnight. Platinum catalyst SIP 6830.3 (1 mg) and dried DMS-H11 (1 g) were added in 50 mL- round-bottom flasks sealed with a rubber septum and a stopcock adapter. The mixture was well-mixed by vigorous shaking. Four reactions under four different atmospheres were created as follows: (a) Dry $N_2$ atmosphere was created by evacuating air from the flask and then backfilling with dry $N_2$ for 4 cycles before the addition of the precursor polymer and the catalyst. (b) Wet $N_2$ atmosphere was created as same as (a) but adding 3 drops of water in the flask and dispersing into the precursor polymer by vigorous shaking. (c) Dry air was created as same as protocol (a) but replacing the dry $N_2$ with dry air. (d) Wet air was created as same as protocol (b) but replacing the dry $N_2$ with dry air. Subsequently, the four flasks were heated at 100 °C for 48 h. The resulting liquid products were analyzed by $^1H$ nuclear magnetic resonance (NMR) and size-exclusion chromatography (SEC).

**Kinetics study**. Two sets of reactions (a) MCR-H21 (2 g) mixed with catalyst SIP6830.3 (2 mg), and (b) MCR-H21 (1 g) mixed with MCR-V21 (1 g) and SIP6830.3 (2 mg) were run in an oven at 100 °C. Samples were taken from the reaction (a) at reaction times of 10 min, 30 min, 60 min, 180 min and 360 min. Samples were taken from reaction (b) at reaction times of 0.5 min, 1 min, 2 min, 5 min, and 30 min. All the samples were cooled down immediately by using dry ice, and analyzed by $^1H$ NMR and SEC.

**Uniaxial tensile test**. Stress–strain responses of elastomers were measured using an Instron 3340 materials testing system (INSTRON, US) at a crosshead speed of 500 mm min⁻¹. Specimens were cut with a dumbbell shape according to ASTM D-638 Type V (width: 3.18 mm; length: 9.53 mm; thickness: 1 mm). Elastic moduli

### Table 4 Information of chemicals used in this study.

| Type of chemical | Chemical structure | Abbreviation | $M_n^a$ (kDa) | $Đ_M^b$ |
|---|---|---|---|---|
| Telechelic Si–H functional PDMS | | DMS-H11 | 1 | 1.3 |
| | | DMS-H21 | 6 | 1.6 |
| | | DMS-H25 | 14 | 1.5 |
| | | DMS-H31 | 24 | 1.5 |
| Mono-Si–H functional PDMS | | MCR-H21 | 7 | 1.1 |
| Multi-Si–H functional PDMS | | HMS-301ᶜ | 2 | – |
| | | HMS-064ᶜ | 60 | – |
| Telechelic vinyl functional PDMS | | DMS-V22 | 8 | 2.1 |
| | | DMS-V25 | 14 | 1.5 |
| | | DMS-V41 | 35 | 2.0 |
| Mono-vinyl functional PDMS | | MCR-V21 | 6 | 1.1 |
| | | MCR-V25 | 23 | 1.3 |

ᵃNumber average molecular weight.
ᵇPoly-dispersity index.
ᶜConcentrations of Si–H groups on HMS-301 and HMS-064 were determined to be 3.74 mol/kg and 0.830 mol kg⁻¹, respectively, based on integrations of $^1H$ spectra.

**Table 5 Formulations of preparing highly stretchable and extremely soft silicone elastomers.**

| Sample | Si-H functional PDMS | | Vinyl functional PDMS | |
|---|---|---|---|---|
| | Mass (g) | Molar amount of Si-H group (mol) | Mass (g) | Molar amount of vinyl group (mol) |
| Ela_DMS-H21_DMS-V22_R1.05 | 4.46 | $1.55 \times 10^{-3}$ | 5.54 | $1.47 \times 10^{-3}$ |
| Ela_DMS-H21_DMS-V22_R1.10 | 4.57 | $1.59 \times 10^{-3}$ | 5.43 | $1.44 \times 10^{-3}$ |
| Ela_DMS-H21_DMS-V22_R1.15 | 4.68 | $1.63 \times 10^{-3}$ | 5.32 | $1.41 \times 10^{-3}$ |
| Ela_DMS-H25_DMS-V25_R1.05 | 5.16 | $3.69 \times 10^{-4}$ | 4.84 | $3.46 \times 10^{-4}$ |
| Ela_HMS_064-MCR-V21_R1.05 | 1.39 | $1.16 \times 10^{-3}$ | 8.61 | $1.10 \times 10^{-3}$ |
| Ela_HMS_064-MCR-V21_R1.20 | 1.56 | $1.30 \times 10^{-3}$ | 8.44 | $1.08 \times 10^{-3}$ |
| Ela_HMS_064-MCR-V21_R1.50 | 1.88 | $1.56 \times 10^{-3}$ | 8.12 | $1.04 \times 10^{-3}$ |
| Ela_HMS_064-MCR-V25_R1.05 | 0.52 | $4.36 \times 10^{-4}$ | 9.48 | $4.15 \times 10^{-4}$ |

were determined by linear fitting of the stress–strain data at a strain range of 0–10%.

**Biaxial tension test**. An elastomer film (1 mm × 50 mm × 50 mm) was marked with a 1 cm-square in the center by using red pigment (PGRED01, 50% in silicone oil, Gelest Inc.). The film was biaxially stretched until it was close to break. The stretched film was placed above a grid pad on a table in order to estimate the size changes of the red square, then it covered the surface of a standard football with a 21 cm-diameter.

**Linear viscoelasticity (LVE)**. Specimen LVE was measured by a strain-controlled rheometer ARES G2 (TA Instruments), using small amplitude oscillatory shear (SAOS). Specimens with a thickness of ∼ mm were cut into cylinders with 8 mm-diameter. The shear strain amplitude was fixed at 1%. For highly stretchable elastomers, frequency sweeps from $1.6 \times 10^1$ to $1.6 \times 10^{-3}$ Hz were conducted at 21 °C and 200 °C, respectively. Time−temperature superposition was used to create master curves based on a reference temperature of 21 °C. For extremely soft elastomers, a frequency sweep was performed from $1.0 \times 10^2$ to $1.0 \times 10^{-2}$ Hz at 21 °C.

**Monitoring curing reactions**. A telechelic hydride precursor polymer (DMS-H11 or DMS-H25, 3 g) and a catalyst (Karstedt´s catalyst or Speier´s catalyst or tris(-dibutylsulfide)rhodium trichloride) were put into an open glass flask (50 mL) and mixed with a magnet stirrer (400 rpm). The flask was heated at controlled temperatures (100 °C or 150 °C) and the samples were taken at given reaction time until a solid elastomer was formed. The storage and loss moduli of the samples at a fixed 2% strain and 0.1 Hz were measured by a strain-controlled rheometer AR-2000 (TA Instruments).

**Time sweep in LVE region**. Modulus complex during curing of a highly stretchable and an extremely soft elastomers were measured by a strain-controlled rheometer AR-2000 (TA Instruments). The mixture of precursor polymers with platinum catalysts was sandwiched by two geometries of instruments. The thickness of the mixture layer is around 0.5 mm and the diameter is the plates is 20 mm. Oscillatory experiments were performed with a controlled temperature of 100 °C, a controlled strain of 1%, and a constant shear frequency of 1 Hz.

**Compression test**. Two pieces of cylinder elastomers (8 mm in diameter) were stacked with a thickness of around 2 mm. Rheometer ARES G2 was used to compress the elastomers by applying two round plates (8 mm in diameter). Applied forces and gaps between plates were recorded during the compression.

**SEC measurement**. SEC was performed on a Tosoh EcoSEC HLC8320GPC instrument equipped with RI and UV detectors and SDV Linear S columns from Polymer Standards Service (PSS). Samples were run in toluene at 35 °C at a rate of 1 mL min$^{-1}$. Molecular weights and $Đ_\text{M}$ were calculated using WinGPC Unity 7.4.0 software and standard linear PDMS were acquired from PSS.

**NMR measurement**. $^1$H NMR spectra of samples dissolved in CDCl$_3$ were performed on a Bruker 600 MHz spectrometer. $^{29}$Si NMR spectra of liquid samples were acquired on a Bruker Avance II spectrometer operating at a magnetic field of 9.4 T ($\nu_\text{L}$ ($^{29}$Si) = 79.495 MHz) and equipped with a 5 mm BBFO probe. A Pi/6 pulse was used for excitation with an interscan delay of 15 seconds. Inverse-gated $^1$H decoupling was applied during acquisition.

Investigated $^1$H decoupling was applied during acquisition. $^1$H-$^{29}$Si long-range correlations were measured using the ASAP-HMB[47] pulse sequence. A 40 ms 10 kHz TOCSY spin-lock period was employed for more efficient relaxation, which greatly reduces the t1-streaks from the methyl signals on chain PDMS. This allows weaker signals in the same $^1$H chemical shift range to be more easily detected. A data matrix of 4096 × 1024 points was acquired (15% NUS in the indirect

dimension) resulting in acquisition times of 340 and 71.5 ms for the direct and indirect dimension, respectively. A repetition delay of 0.750 s was used, i.e. a total interscan delay of 1.09 s, and 8 scans was measured for each FID. The transfer delay was optimized for detection of long-range couplings of 7 Hz. Omitting a low-pass J-filter after the spin-lock period allowed for the $^1$J$_\text{H-Si}$ coupling from the hydride species to be detected.

$^{29}$Si solid-state NMR MAS spectra of investigated elastomers were acquired on a Bruker Avance III HD spectrometer operating at a magnetic field of 14.05 T ($\nu_\text{L}$($^{29}$Si) = 119.2 MHz) and equipped with a 4 mm CP/MAS broadband probe. The spectra were acquired with a spinning frequency of 6 kHz, a $\pi$/2 pulse of 4.75 ms, an acquisition time of 35 ms and 10 seconds of interscan delay. This was determined to be sufficient for full relaxation of the two observed signals for elastomers. High-power $^1$H SPINAL64 decoupling ($\nu_\text{RF}$ = 100 kHz) was employed during acquisition. The prepared elastomers were cut into smaller pieces and packed in 4 mm o.d. zirconia rotors. Chemical shifts are reported relative to TMS (0.0 ppm). NMR spectra were analyzed using MestReNova-11.

**Swelling experiment**. Gel fraction amounts of elastomers were determined by swelling experiments. A roughly 0.1 g sample (mass = $m_\text{i}$, determined to 4 significant digits) with dimensions of ~10 mm × 10 mm × 1 mm was immersed in around 20 mL of chloroform at room temperature. The chloroform was replaced after 1 day and decanted off after 2 days. The swollen sample (mass = $m_\text{s}$) was then washed with fresh chloroform and dried for 24 h at room temperature under ambient pressure (mass = $m_\text{d}$). The weight of $m_\text{i}$, $m_\text{s}$, and $m_\text{d}$ were recorded. The gel fraction (%) was calculated as $\frac{m_\text{d}}{m_\text{i}} \times 100\%$, and the swelling ratio was calculated as $\frac{m_\text{s}-m_\text{d}}{m_\text{d}}$. Each sample was triplicated.

## Data availability

All data are available in the manuscript or Supplementary Information. Source data are provided with this paper.

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

## Acknowledgements

Pengpeng Hu acknowledges the financial support from China Scholarship Council (CSC, 201804910748, PH) and Department of Chemical and Biochemical Engineering, Technical University of Denmark. We appreciate the measurements of NMR spectra at NMR Center at DTU Chemistry

## Author contributions

P.P.H. designed and conducted the experiments, analysis the data, drafted and revised the paper. A.L.S. was responsible for the overall research direction and objectives, supervised the project, and revised the paper. J.M. supervised the project, and revised the paper.

## Competing interests

The authors declare no competing interests.
