## [Peer Review File · Nature Communications]

REVIEWER COMMENTS

Reviewer #1 (Remarks to the Author):

This is a well prepared and clear report of careful preparative chemistry and sufficient product characterization. Although the catena-crosslinked silicones have been known since 2016 (reference 15), there is little in the literature concerning making use of this crosslinking mechanism and this paper will open eyes concerning it's usefulness.

I don't have any criticism of the document and suggest publishing it "as is."

Reviewer #2 (Remarks to the Author):

Skov and coworkers have developed a new curing method to prepare highly stretchable silicone elastomers as well as extremely soft silicone elastomers. The focus of this paper is the use of a side reaction in conventional formulations as the main reaction. This original approach allows the preparation of various networks using commercial products in one pot.

It is a well-written paper but I don't understand the choice of the journal. I think this work is more suitable for a journal with a strong focus on polymer and chemistry such as Progress in Polymer Science.

Prior to publication, I join some minor points that need to be addressed:

-This side reaction (the reaction of silicon hydride with atmospheric moisture or oxygen) is well known in the literature. I think the authors should state it more precisely. Some references about it: Quan, X. "Properties of post-cured siloxane networks." *Polymer Engineering & Science* 29.20 (1989): 1419-1425. Or Esteves, A. C. C., et al. "Influence of cross-linker concentration on the cross-linking of PDMS and the network structures formed." *Polymer* 50.16 (2009): 3955-3966.)

-Does the author have the ²⁹Si solid-state NMR spectra of the reaction products after heating at 100°C for 48h under dry N₂, wet N₂ and dry air conditions?

-The mechanism proposed in Figure 2.c is quite unusual and needs more scientific proofs. Furthermore, I don't think that the snail-shape is suitable for easy reading.

- The authors assigned this reaction to be « easy curing ». However, this reaction takes 48 h @ 100 °C which is very long compared to current silicone curing. Is it possible to improve this by increasing the temperature or adding more/other catalyst?

-The reaction progression is proposed in Figure 3. I think the authors should add in the legend how they obtained this (i.e. NMR). Did the authors test FT-IR and/or Raman? I think it would help them to understand more the mechanism.

-Did the authors tested the gels fractions of obtained networks? I am curious about the real gel fraction of some low-crosslinked networks.

-As the reaction takes 48h @ 100°C, did the authors think about exchange reactions?

Dear editor, Dear reviewers,

The authors would like to thank the reviewers for their insightful comments and constructive suggestions, and thank Nature Communications for providing the opportunity to re-submit the revised manuscript. The article has been carefully revised according to the suggestions and comments from the reviewers, and the requirements for re-submission from the journal. The main changes are:

- Mechanism study of curing reactions and the proposed mechanism (Fig. 2 a, b, and c and; the corresponding descriptions in the main text) were updated.
- Effects of the curing conditions on the curing rates were added in the manuscript.
- Swelling properties of the investigated elastomers, including gel fractions and swelling ratios, were shown in the manuscript.
- Manuscript was formatted according to the Guide to Formatting Articles.
- Source data file for the Figures and tables except the NMR spectra are provided.
- Chemical identity based on the NMR spectra was added in the supporting file.

Below we have included the detailed responses on a point-by-point basis. We have tracked the changes in the main document.

Sincerely,

Anne Ladegaard Skov/ On behalf of all authors

Responses to the comments from reviewer 2

Note: The figures and table below are shown in the revised manuscript with the same numbering. The mentioned numbers of lines are based on the revised manuscript without showing traces of changes.

Comment 1 *This side reaction (the reaction of silicon hydride with atmospheric moisture or oxygen) is well known in the literature. I think the authors should state it more precisely. Some references about it: Quan, X. "Properties of post-cured siloxane networks." Polymer Engineering & Science 29.20 (1989): 1419-1425. Or Esteves, A. C. C., et al. "Influence of cross-linker concentration on the cross-linking of PDMS and the network structures formed." Polymer 50.16 (2009): 3955-3966.).*

Response 1 Thanks for the suggestion and the excellent references. We agree that these references are highly relevant and therefore we replaced "Hydrosilanes readily undergo hydrolysis and alcoholysis reactions with water and alcohols, respectively, under basic or strongly acidic conditions or in the presence of radicals, metals, or transition metal complexes.^{25,26} In the presence of a platinum catalyst and water, telechelic Si-H functional PDMS can thus be hydrolyzed into Si-OH, which may further undergo condensation to form extended chains." with "As well-known side reactions of the conventional curing reaction (Fig. 1a), hydrolysis of hydrosilanes with atmospheric moisture, and oxidation of hydrosilanes with atmospheric oxygen both lead to the formation of Si-OH groups^{25,26}. The Si-OH groups can form Si-O-Si by reacting with themselves or with Si-H groups^{27,28}. In the presence of a platinum catalyst, moisture, and oxygen, telechelic Si-H functional PDMS can thus be transferred into telechelic Si-OH, which may further undergo condensation to form extended chains." at lines 65-71 of the revised manuscript, where the two suggested references are now reference 25 and 26.

Comment 2 *Does the author have the 29Si solid-state NMR spectra of the reaction products after heating at 100°C for 48h under dry N2, wet N2 and dry air conditions?*

Response 2 We have repeated the oxygen and water controlling reactions of DMS-H11. Solid state NMR spectra refer to the NMR spectra of solids and semi-solids. Since the reaction products after heating at 100 °C for 48 h under dry N₂, wet N₂, and dry air conditions are liquids, we dissolved the liquid samples in CDCl₃ and measure their ¹H NMR and ²⁹Si NMR spectra (**Fig. 2a and b**). Previous ²⁹Si solid state NMR spectrum of the resulting elastomer from DMS-H11 is also shown in **Fig. 2b**. 2D NMR spectrum of the liquid product from dry air condition was measured in order to figure out the correlation between H atom and Si atom (**Supplementary Fig. 2**).

Based on the new results (**Fig. 2b** and **Supplementary Fig. 2**), we found CH₃SiHO₂ and (CH₃)₂CH₂SiO on the samples from wet N₂ and dry air conditions according to their ²⁹Si NMR and 2D NMR (¹H-²⁹Si coupling) spectra. These new structures eventually contribute to crosslinking after full conversion of Si-H functional groups.

The new observations are consistent with our previous results and proposed mechanism, but provide additional details on the curing reactions: Firstly, CH₃SiHO₂ is regarded as an intermediate structure during the curing reaction, which is eventually transformed into CH₃SiO₃ (see Ela_DMS-H11 in **Fig. 2b**) after curing. This intermediate structure was not observed in the previous manuscript. Secondly, water plays the same role as oxygen to produce CH₃SiHO₂ and (CH₃)₂CH₂SiO as the two structures are

observed on the ^{29}Si spectra of the samples from dry air and wet N_2 conditions; while we previously emphasized the oxidation of Si-H only with oxygen. Finally, due to the observation of CH_3SiHO_2 , carbon radical ($\text{SiCH}_2\cdot$) is regarded to be generated from the methyl groups at the termini of the PDMS chains. Further oxidation of $\text{SiCH}_2\cdot$ leads to the creation of $\text{SiCH}_2\text{OO}\cdot$, which enables the formation of CH_3SiHO_2 . (Fig. 2c-I, J, and K); while, in the previous version of the manuscript, the carbon radical was regarded to be generated on arbitrary methyl groups along the PDMS chains.

Fig. 2 and **Supplementary Fig. 2**, and the corresponding descriptions at lines 81-138 in text are updated in the revised manuscript.

Fig. 2(a) ^1H NMR spectra and (b) ^{29}Si NMR spectra of a telechelic Si-H functional PDMS (DMS-H11) and its reaction products after heating at 100°C for 48 h under dry N_2 , wet N_2 , and dry air conditions, respectively. Additionally, ^{29}Si solid state NMR spectrum of the elastomer (Ela_DMS-H11) cured under normal air atmosphere is shown in (b).

Supplementary Fig. 2. 2D NMR spectrum (^1H - ^{29}Si NMR coupling) of DMS-H11's reaction product after heating at 100°C for 48 h under dry air condition.

Comment 3 The mechanism proposed in Fig. 2.c is quite unusual and needs more scientific proofs. Furthermore, I don't think that the snail-shape is suitable for easy reading?

Response 3 The proposed mechanism in Fig. 2c in the manuscript is based on reported literature references (reference 24, 29-37) and our experimental observations. Important and recent literature agree with the mechanism, including the article authored by Mike Brook et al (reference 24), where the authors propose the same radical groups (radical 1, 2, and 3 in Fig. 2) during the curing of telechelic hydride functional PDMS with no catalyst at 200 °C. Also supported by reported literature (references 24, 36, and 37), the proposed radicals are able to combine to form stable molecular structures (SiOCH_2Si , $(\text{CH}_3)_2\text{CH}_2\text{SiO}$) and CH_3SiO_3 , which are consistent with our observations from ^1H NMR and ^{29}Si NMR spectra.

We changed the snail-shape of Fig. 2c in order to make it suitable for easy reading.

Comment 4 The authors assigned this reaction to be « easy curing ». However, this reaction takes 48 h @ 100 °C which is very long compared to current silicone curing. Is it possible to improve this by increasing the temperature or adding more/other catalyst?

Response 4 Thank you for the highly relevant question about the curing rate. The conditions of 100 °C and 48 h are specifically for the curing reaction of precursor polymer DMS-H11 under controlled moisture and oxygen in a sealed flask, where the concentrations of water and oxygen decrease during the reaction. Therefore, the curing conditions of 100 °C and 48 h are not representative for general curing reactions under normal air where the water and oxygen concentrations are almost constants during curing.

During this revision, we studied the effects of Si-H concentration of telechelic Si-H functional PDMS, reaction temperatures, concentration of catalyst, and types of catalyst on the curing reaction under a normal air condition. The reactions took place in glass flasks with a magnet stirrer stirring and the samples were transferred from the flasks to a rheometer for measuring their storage and loss moduli after given reaction times. During the curing reactions, the precursor polymer chains are gradually connected into 3-dimensional networks, and as result, the storage and loss moduli increase. However, this method allows only to monitor the reactions before gelations, as shown in **Supplementary Fig. 3a** and **b**. The times when the precursor polymers under various conditions become solids were recorded, and shown in **Supplementary Fig. 3c**.

Karstedt's catalyst, Speier's catalyst, and tris(dibutylsulfide)rhodium trichloride catalyst have been widely applied as effective hydrosilylation catalysts (*RSC Adv.*, 2015, 5, 20603). Tris(pentafluorophenyl)borane catalyst is an efficient catalyst for Piers-Rubinsztajn reaction, in which the catalyst activates Si-H functional groups (*Chem. Eur.J.* 2018, 24, 8458 –8469). Therefore, these catalysts were selected as candidate catalysts for our curing reaction. As **Supplementary Fig. 3c** shows, Karstedt's catalyst, Speier's catalyst and tris(dibutylsulfide)rhodium trichloride catalyst enable solid elastomers from precursor polymer DMS-H25 after reactions at 100 °C for 3 h, 8 h, and 12 h, respectively, while tris(pentafluorophenyl)borane catalyst results in liquid products only, even after reaction for 48 h. It is speculated that solely extended chains are formed via hydrolysis and condensation of Si-H functional groups in the presence of the tris(pentafluorophenyl)borane catalyst.

As **Supplementary Fig. 3b** and **c** shows, lower hydride concentration of precursor polymer, higher temperature, and higher platinum catalyst concentration significantly increase the curing rates, when Karstedt's catalyst was used. Particularly, a solid elastomer was formed after only 0.5 h at 150 °C in the presence of 30 ppm Karstedt's catalyst, and a solid elastomer was formed after 1 h at 100 °C, in the presence of 150 ppm Karstedt's catalyst. However, the resulting elastomer became brown when using 150 ppm catalyst concentration, which is a general issue for conventional hydrosilylation curing reactions (*RSC Adv.*,2015, 5, 20603).

The corresponding experimental procedure and the results were added in the supporting information (Monitoring curing reactions, **Supplementary Fig. 3a, b,** and **c**) in the revised manuscript. The description of the results "*The platinum catalysts (Karstedt's catalyst and Speier's catalyst) and tris(dibutylsulfide)rhodium trichloride catalyst are shown as effective catalysts for the curing of telechelic Si-H functional DPMS (Supplementary Fig. 3c). The curing rate can be improved by applying a lower hydride concentration of precursor polymer, higher temperature, and higher platinum catalyst concentration (Supplementary Fig. 3b and c). Particularly, a solid elastomer was formed from DMS-H25 ($M_n=14$ kDa, 0.14 mol/kg Si-H functional groups) after only 0.5 h under conditions of 150 °C and 30 ppm Karstedt's catalyst.*" was added at lines 150-156 of the revised manuscript.

Supplementary Fig. 3 Curing rates for telechelic hydride functional PDMS under various conditions. (a) and (b): Storage moduli (G') and loss moduli (G'') of reaction products over reaction time. The samples were sheared at 0.1 Hz with a fixed 2% strain. Specifically, (a) Reactions of precursor polymer DMS-H25 are catalyzed at 100 °C by various catalysts with 30 ppm concentration, i. e. Karstedt's catalyst (Kar.), Speiers's catalyst (Spe.), tris(pentafluorophenyl)borane catalyst (Bor.), and tris(dibutylsulfide) rhodium trichloride catalyst (Rho.). (b) The reactions are catalyzed by Karstedt's catalyst under different Si-H concentrations (2.0 mol/kg Si-H in precursor DMS-H11 and 0.14 mol/kg Si-H in precursor DMS-H25), reaction temperatures (100 °C and 150 °C) and catalyst concentrations (6 ppm, 30 ppm and 150 ppm). (c) Solidification times for telechelic hydride functional PDMS under various conditions.

Comment 5 *The reaction progression is proposed in Figure 3. I think the authors should add in the legend how they obtained this (i.e. NMR). Did the authors test FT-IR and/or Raman? I think it would help them to understand more the mechanism?*

Response 5 Thanks for the suggestion. In order to describe how we obtain the reaction progression, "*The extents of reaction were determined based on ^1H NMR spectra (Supplementary Fig. 4)*" was added in the legend of Fig. 3 in the revised manuscript.

Changes in chemical compositions has been detected using Raman microscopy (CRM), Fourier-transform infrared spectroscopy (FTIR), and NMR spectroscopy. According to the two reference articles mentioned in question 1 by the reviewer, *in situ* confocal CRM and ATR-FTIR spectroscopy were used to monitor the consumption of Si-H and C=C bonds simultaneously during the curing reaction. In our case, as we present in Fig. 3 of the manuscript, the reaction progression was monitored based on the consumption of Si-H groups by ^1H NMR, which generally possesses comparable detection sensitivity to CRM and ATR-FTIR.

We tested the versatility of FTIR spectroscopy (Nicolet iS50 FTIR Spectrometer, Thermo Fisher Scientific) for DMS-H11 and the resulting elastomer. As the spectra in the **Supplementary Fig. 7** show, the peaks assigned to the Si-H functional group at wavenumbers 2126 and 909 cm^{-1} almost disappeared, indicating the efficient consumption of the Si-H functional group after the curing reaction (*Polymer* 50 (2009) 3955-3966). No new peaks were observed on the FTIR spectra of the elastomers. Thus, we agree that FT-IR or Raman may also be used to monitor the reaction progression, but based on the presented IR spectra, we are not convinced that using these techniques adds anything in terms of understanding the mechanism.

Supplementary Fig. 7. Fourier-transform infrared spectroscopy (FTIR) of elastomers (Ela_DMS-H11 and Ela_DMS-H21) and a precursor polymer (DMS-H11)

Comment 6 Did the authors tested the gels fractions of obtained networks? I am curious about the real gel fraction of some low-crosslinked networks?

Response 6 Thanks for the valuable suggestion. We tested swelling properties (gel fractions and swelling ratios) of the networks in chloroform. Overall, the gel fractions of the highly stretchable elastomers and the extremely soft elastomers are 52-83%, which are lower than the value of 97% obtained for the conventional elastomer (Ref_DMS-V25). The higher stretchability or the lower softness, the lower gel fraction of the elastomer. The elastomer with the lowest softness (Ela_HMS-064_MCR-V25_R1.05) has a lowest gel fraction. The gel fractions of the extremely soft elastomers (the prepared bottle-brush elastomers) are similar with those of the recently reported bottle-brush elastomers with the similar softness (*J. Am. Chem. Soc.* 2021, 143, 9866–9871, *Mater. Horiz.*, 2020, 7,181–187, *Adv. Mater.* 2015, 27, 5132–5140). The highly stretchable elastomers show the highest swelling ratio of 128, which is around 10 times larger than the conventional elastomer (Ref_DMS-V25). The extremely high swelling ratio is contributed to the highly flexible nature of the elastomer in which entanglements greatly outnumber crosslinks.

The main text of the revised manuscript has been modified: “*Swelling experiments show 67-83% gel fractions of the elastomers (Supplementary Table 5). The most stretchable elastomer (Ela DMS-H25 DMS-V25 R1.05) exhibits the highest swelling ratio of 128, which is around 10 times larger than the conventional elastomer (Ref DMS-V25). The extremely high swelling ratio correlates with the high flexibility of the elastomer where entanglements greatly outnumber crosslinks.*” were added at lines 233-237 in the revised manuscript. Besides, the sentence at lines 257-260 “*The large differences between M_c and M_{c-SiH} can be explained by the preferentially intramolecular reactions of the multi-functional bottle-*

*brush chains, which result in a large fraction of elastically inactive loops and dangling⁴⁷” was replaced by “The large differences between M_c and M_{c_SiH} can be explained by the preferentially intramolecular reactions of the multi-functional bottle-brush chains, which result in a large fraction of elastically inactive loops and dangling chains⁴⁷. This is illustrated by the lower gel fractions (52-83 %) compared to the conventional elastomer (97 %), see **supplementary Table 5).**”*

Table Gel fractions and swelling ratios of the investigated networks

Samples	Gel fractions (%)	Swelling ratio
Ela_DMS-H21_DMS-V22_R1.05	67±2	98±13
Ela_DMS-H21_DMS-V22_R1.10	73±1	47±4
Ela_DMS-H21_DMS-V22_R1.15	83±1	26±4
Ela_DMS-H25_DMS-V25_R1.05	69±1	128±17
Ela_HMS-064_MCR-V21_R1.05	73±1	30±2
Ela_HMS-064_MCR-V21_R1.20	78±1	20±1
Ela_HMS-064_MCR-V21_R1.50	83±1	15±3
Ela_HMS-064_MCR-V25_R1.05	52±1	48±2
Ela_DMS-H21	93±1	9±1
Ela_DMS-H25	93±1	12±2
Ela_DMS-H31	88±1	14±2
Ela_HMS-064	88±1	13±2
Ref_DMS-V25	97±0	12±1

Comment 7 *As the reaction takes 48h @ 100°C, did the authors think about exchange reactions?*

Response 7 As it is mentioned in the response to comment 4, curing for 48 h at 100 °C is specific for DMS-H11 under a controlled atmosphere of moisture and oxygen, which is not representative of the general curing reaction of telechelic hydride terminated PDMS. The curing process can be accelerated significantly by increasing temperature, catalyst concentration and by decreasing the concentration of the Si-H group of precursor PDMS allowing the formation of elastomers in less than one hour (see **supplementary Fig. 3**).

We cannot exclude the existence of other reactions in the process. However, we are not entirely sure what kind of exchange reactions the reviewer is proposing and would appreciate additional details.

REVIEWERS' COMMENTS

Reviewer #2 (Remarks to the Author):

I would like to thank the authors for this revision. All of my questions were answered with clarity and quality.

Upon reflection, I felt that this paper is appropriate for this journal.